# Association between the stage of labour during caesarean delivery with adverse maternal and neonatal outcomes among referred mothers to tertiary centres in resource-limited settings

Dereje Zewdu [ID],[1] Temesgen Tantu,[2] Fikretsion Degemu,[3] Mukerem Abdlwehab[3]

[1]Department of Anesthesia, Wolkite University, Welkite, Ethiopia
[2]Obstetrics and Gynecology, Wolkite University, Welkite, Ethiopia
[3]Department of Pediatrics and Child Health, Wolkite University, Welkite, Ethiopia

**Correspondence to**
Dr Dereje Zewdu;
derejezewdu1529@gmail.com

## ABSTRACT

**Objective** Although the caesarean delivery (CD) rate has substantially increased, little is known about its impacts when performed in the first and second stages of labour on fetomaternal outcomes, especially among referred mothers. Thus, this study aimed to investigate the association between CDs performed during the first and second stages of labour and poor maternal and neonatal outcomes among mothers referred to tertiary centres.

**Setting** This retrospective cohort study analysed medical records of mother–infant pairs from September 2020 to May 2023 in Southern Ethiopia.

**Participants** We retrospectively collected data from 848 participants who underwent emergency CD on a referral basis during the study period.

**Primary outcome measure** The primary outcomes of interest were adverse maternal and neonatal outcomes. Data were analysed using descriptive and inferential statistics.

**Results** Of the 848 CDs, 722 (85.2%) and 126 (14.8%) were performed at the first and second stages of labour, respectively. Caesarean sections performed at the second stage were higher with nulliparity, increased maternal age, and birth weight. Compared with the first-stage CD, the second-stage CD was associated with a significantly increased risk of adverse maternal (OR 3.7, 95% CI 2.4 to 5.7) and neonatal outcomes (OR 2.0; 95% CI 1.3 to 2.9), including neonatal death.

**Conclusion** Second-stage CDs have an increased risk of adverse maternal and neonatal outcomes. Strengthening and improving obstetric emergency surgical services and intensive neonatal care for those populations would help decrease the maternal and fetal negative consequences.

## INTRODUCTION

The caesarean delivery (CD) rate has significantly increased globally, representing one-third of total institutional deliveries in Ethiopia.[1 2] Although CD is generally a lifesaving procedure, it may lead to increased maternal and neonatal complications and healthcare expenses.[3 4] In many low-income countries where equitable access to obstetric

## STRENGTHS AND LIMITATIONS OF THIS STUDY

⇒ The study evaluated the impact of second-stage caesarean delivery on fetomaternal outcomes among referred mothers using a larger sample size.
⇒ The study followed the maternal and neonatal characteristics from admission to hospital discharge.
⇒ The study included 3-year data, which is relatively representative of the existing problems.
⇒ The retrospective nature of the study design may result in variable selection bias.
⇒ Extrapolating the findings to non-referred participants might be difficult.

surgical services is limited, the patient frequently gets referred to nearby tertiary centres and receives delayed treatment, resulting in compromised maternal and neonatal outcomes.[5 6]

It is interesting to note that increased access to CD in resource-limited settings could help reduce the incidence of maternal and newborn morbidity and mortality.[7] Nonetheless, being referred to other centres for CD and delays in timely intervention contribute to the likelihood of unfavourable fetomaternal outcomes.[8] Despite these, the impact of the stage of labour during CD on maternal and neonatal adverse outcomes among those populations has not been well established. While clinicians carried out many CD procedures during the first stage of labour with fewer undesirable outcomes for both the mother and the newborn, the second-stage CD is technically more difficult and poses serious consequences.[9 10]

Previous studies demonstrated inconclusive and conflicting results on the association between CD at the stage of labour and fetomaternal outcomes. Some studies have found a significant correlation between CD performed at the second stage of labour and

adverse maternal effects.[11–18] On the contrary, others found no association between second-stage CD and poor maternal outcomes.[19–21] Poor maternal outcomes among CD performed at the second stage of labour include increased blood loss, uterine atony, need for blood transfusion and perioperative complications.

Concerning neonatal outcomes, one study reported serious neonatal complications are common in the CD performed at the first stage of labour.[10] Nevertheless, other studies found higher poor outcomes in second-stage CD[13 14 21 22] and no association between these measurements.[15 17 23–25] Neonatal adverse outcomes include a low fifth min Apgar score, increased need for neonatal intensive care unit (NICU) admission and neonatal death.

As far as our search, no other studies have compared the association between CD performed at the first and second stage of labour on their neonatal and maternal outcomes among referred mothers to tertiary centres. For practitioners to pinpoint the existing clinical gap and use the findings to improve clinical practice, it would be beneficial to understand the connections between CD carried out at the first and second stages of labour and undesirable newborn and mother outcomes.

Therefore, we aimed to investigate the association between CD performed at the first and second stage of labour with adverse maternal and neonatal outcomes among mothers referred to Wolkite University Referral and Teaching Hospital in Southern Ethiopia. The findings of this study provide information for clinicians, researchers and policy-makers to maximise the quality of perioperative obstetric and intensive neonatal care and decrease second-stage CD-related fetomaternal complications.

## METHODS

### Study population and study design

This retrospective cohort study included all consecutive women who underwent primary CD for singleton term pregnancy between September 2020 and May 2023 at Wolkite University Specialized and Teaching Hospital in Ethiopia.

We included a total of 848 consecutive primary CD performed for referred mothers during the study period for final analysis. Exclusion criteria were as follows: mothers with multiple gestation, preterm birth, intra-uterine growth restriction, fetal anomalies and incomplete documentation.

### Data collection

We retrospectively collected data from the maternal and neonatal medical records, including maternal and neonatal baseline characteristics, perioperative data, and adverse maternal and neonatal outcomes of all study participants using a checklist adapted from prior studies.[26]

The maternal and neonatal baseline characteristics included maternal age, body mass index, parity, residency area, educational status, onset and duration of labour, gestational age, birth weight and fetal sex. Perioperative maternal and neonatal characteristics included skin incision to delivery time, operation duration, estimated blood loss, uterine atony, unintentional uterine tear extension, uterine artery ligation, need for blood transfusion, ureteral/bowel injury, perioperative hysterectomy, need for intensive care unit (ICU) admission and maternal death, as well as fifth min Apgar score, need for NICU admission and neonatal death.

### Study outcomes

The primary outcomes of interest were adverse maternal and neonatal outcomes. The maternal outcomes included one or more of the following: perioperative haemorrhage, uterine artery ligation, unintentional uterine extension, ureteral or bowel injury, peripartum hysterectomy and maternal death. Neonatal adverse outcomes included one or more of the following: fifth min Apgar score <7, need for NICU admission and neonatal death.

### Statistical analysis

We entered, coded and analysed the data using SPSS (SPSS for Windows V.26.0, SPSS). We used the Student's t-test test for numerical variables and described them as mean±SD.

We presented the categorical data as frequency and percentage, and the association between independent and dependent variables was analysed using the $\chi^2$ and Fisher's exact tests for statistical differences between groups. Logistic regression analysis examined the impacts of CD performed at the first and second stages of labour on maternal and neonatal adverse outcomes. Results displayed as an OR with a 95% CI and a p<0.05 are considered statistically significant.

### Patient and public involvement

None.

## RESULTS

### Maternal and neonatal baseline characteristics

Of 897 women who underwent primary caesarean section performed during the study period, about 848 (94.5%) were included for final analysis. Of these, 722 (85.1%) CDs were in the first stage and 126 ((14.9%) in the second stage of labour. While education, referral status, the labour onset and fetal sex between CD performed at the first and second stages were comparable, nulliparity, the onset of labour to delivery time, and birth weight were significantly higher in the second-stage CD 58.7% vs 35.7%; p<0.001), (14.8±10.4 vs 10.6±7.1; p<0.001), and (3298.1±338.3 vs 3444.1±322.3; p<0.001), respectively, as shown (table 1).

### Perioperative maternal and neonatal characteristics

Maternal perioperative characteristics, as shown in (table 2), the CD performed at the second stage of labour was significantly associated with prolonged skin incision-to-delivery time, operation time, postpartum

**Table 1** Maternal and neonatal baseline characteristics between CD performed at the first and second stages of labour

| Variables | CD at first stage of labour (n=722) | CD at second stage of labour (n=126) | P value |
|---|---|---|---|
| Maternal age, year, mean±SD | 26.5±4.8 | 24.9±4.5 | **<0.001** |
| BMI, Kg/m$^2$, mean±SD | 27.9±2.5 | 28.6±2.5 | 0.004 |
| Nulliparity, n (%) | | | **<0.001** |
| Yes | 258 (35.7) | 74 (58.7) | |
| No | 464 (64.3) | 52 (41.3) | |
| Educational status, n (%) | | | 0.17 |
| Literate | 588 (81.4) | 96 (76.2) | |
| Illiterate | 134 (18.6) | 30 (23.8) | |
| Referral status, n (%) | | | 0.62 |
| Rural | 338 (46.8) | 62 (49.2) | |
| Urban | 384 (53.2) | 64 (50.8) | |
| The onset of labour, n (%) | | | 0.65 |
| Spontaneous | 662 (91.7) | 114 (90.5) | |
| Induced | 60 (8.3) | 12 (9.5) | |
| Labour onset to delivery, hours, mean±SD | 10.6±7.1 | 14.8±10.4 | **<0.001** |
| Neonatal | | | |
| GA at delivery, weeks, mean±SD | 38.6±1.3 | 38.7±1.5 | 0.54 |
| Birth weight at delivery, gram, mean±SD | 3298.1±338.3 | 3444.1±322.3 | **<0.001** |
| Fetal sex, n (%) | | | 0.9 |
| Male | 375 (51.9) | 66 52.4) | |
| Female | 347 (48.1) | 60 (47.6) | |

A p<0.05 is considered significant.
The values in bold are statistically highly significant.
BMI, body mass index; CD, caesarean delivery; GA, gestational age.

haemorrhage, uterine atony, uterine artery ligation, need for blood transfusion, unintentional uterine tear extension, perioperative hysterectomy, need for ICU admission and even ureteral/bowel injury (p<0.001, p<0.001, p<0.001, p<0.001, p=0.002, p=0.008, p<0.001, p=0.016, p=0.026, p=0.036, respectively).

Similarly, neonatal outcomes, including fifth min Apgar score <7, need for NICU admission and neonatal death, were significantly higher in CD performed at the second stage than in the first stage of labour (22.2% vs 8.3%; p<0.001, 34.9% vs 24.6%; p<0.001, 14.3% vs 3.6%; p<0.001), respectively, as shown in (table 2).

### Association between CD performed at the first and second stage of labour with adverse maternal and neonatal outcomes

Logistic regression analysis showed that the likelihood of developing adverse maternal outcomes after CD performed at the second stage of labour was 3.7 times (OR 3.7, 95% CI 2.4 to 5.7) higher than the first-stage CD. Similarly, second-stage CD increases the risk of developing adverse neonatal outcomes twofold (OR 2.0; 95% CI 1.3 to 2.9) compared with CD performed at the first stage of labour, as shown in table 3.

## DISCUSSION

In this retrospective study involving 848 women who underwent emergency CD at a referral hospital in Southern Ethiopia, we compared maternal and neonatal adverse outcomes of CD performed during the first and second stages of labour. Our main findings were that CD performed at the second stage was associated with an increased risk of maternal and neonatal adverse outcomes compared with CD performed at the first stage of labour.

Increased adverse maternal outcomes included longer operation times, estimated blood loss or postpartum haemorrhage, uterine atony and uterine artery ligation procedures, suggesting that CD at the second stage might be technically challenging and associated with higher odds of complications.[9 10] This is likely due to the engagement of the fetal head in the pelvis, the difficulty in delineating the bladder, and the larger infants that are perhaps more difficult to remove from the uterus.[11 12]

The increased amount of blood loss because of uterine atony and unintentional uterine extensions is frequent in the second stage CD, as revealed in the present and previous studies.[13 14] Other studies also showed that unintended uterine extension was associated with longer

**Table 2** Perioperative maternal and neonatal characteristics between CD performed at the first and second stages of labour

| Variables | CD at the first stage of labour (n=722) | CD at the second stage of labour (n=126) | P value |
|---|---|---|---|
| Skin incision to delivery time, minutes | 6.1±1.6 | 7.7±2.2 | **<0.001** |
| Duration of operation, minutes | 41.8±20.1 | 49.4±26.1 | **<0.001** |
| Duration of operation >1 hour, n (%) | | | **<0.001** |
| Yes | 122 (16.9) | 42 (33.4) | |
| No | 600 (83.1) | 84 (66.6) | |
| Estimated blood loss, mL | 464.5±342.2 | 599.2±539.2 | **<0.001** |
| Perioperative haemorrhage (> 1000 mL), n (%) | | | **<0.001** |
| Yes | 44 (6.1) | 20 (15.9) | |
| No | 678 (93.9) | 106 (84.1) | |
| Uterine atony, n (%) | | | **<0.001** |
| Yes | 34 (4.7) | 18 (14.3) | |
| No | 688 (95.3) | 108 (85.7) | |
| Uterine artery ligation, n (%) | 18 (2.5) | 10 (7.9) | 0.002 |
| Blood transfusion, n (%) | | | 0.008 |
| Yes | 22 (3) | 10 (7.9) | |
| No | 700 (97) | 116 (92.1) | |
| Unintentional uterine tear extension, n (%) | 4 (0.5) | 7 (5.7) | **<0.001** |
| Perioperative hysterectomy, n (%) | 8 (1.1) | 5 (4.1) | 0.016 |
| ICU admission, n (%) | 12 (1.6) | 6 (4.9) | 0.026 |
| Ureteral/bowel injury, n (%) | 4 (0.5) | 3 (2.4) | 0.036[*] |
| Maternal death, n (%) | 3 (0.4) | 2 (1.6) | 0.08 |
| Neonatal | | | |
| Fifth minute Apgar score <7, n (%) | | | **<0.001** |
| Yes | 60 (8.3) | 28 (22.2) | |
| No | 662 (91.7) | 98 (77.8) | |
| Need for NICU admission, n (%) | | | **<0.001** |
| Yes | 178 (24.6) | 44 (34.9) | |
| No | 544 (75.4) | 82 (65.1) | |
| Neonatal death, n (%) | | | **<0.001** |
| Yes | 26 (3.6) | 18 (14.3) | |
| No | 696 (96.4) | 108 (85.7) | |

The values in bold are statistically highly significant.
*Fisher's exact test.
CD, caesarean delivery; ICU, intensive care unit; NICU, neonatal intensive care unit.

operation time, more blood loss, postpartum anaemia, and an increased need for blood transfusion.[15 16]

We believe that the extended labour duration and more substantial uterine fatigue may be the reason for the increase in uterine atony incidence in the second stage CD.[17] Our study also showed because of the increased amount of blood loss and uterine atony, an increased need for blood transfusion and uterine artery ligation procedures are more frequent in second-stage CD, in line with previous reports.[18] Therefore, it is crucial to consider effective uterotonic agents, leveraging the expertise of experienced obstetricians and continual monitoring to lessen maternal morbidity in these scenarios.

Contradicting our findings, a few studies[19–21] reported no association between second-stage CD and more blood loss, uterine atony, and other poor maternal outcomes. These discrepancies may occur because of the variations in the study population recruited for analysis (urgency of the surgery, indications for CD and delayed presentation in their referral status), fluctuations in sample size and study design, and disparities in the standard of obstetric and neonatal care between the

**Table 3** Association of adverse maternal and neonatal outcomes with CD performed at the first and second stages of labour

| Outcomes | CD at the first stage of labour (n=722) | CD at the second stage of labour (n=126) | ORs (95% CI) | P value |
|---|---|---|---|---|
| Adverse maternal outcomes | | | | **<0.001** |
| Yes | 88 (12.2) | 43 (34.1) | 3.7 (2.4 to 5.7) | |
| No | 634 (87.8) | 88 (65.9) | 1 | |
| Adverse neonatal outcomes | | | | **<0.001** |
| Yes | 198 (27.4) | 54 (42.8) | 2.0 (1.3 to 2.9) | |
| No | 524 (72.6) | 72 (57.2) | 1 | |

The values in bold are statistically highly significant.
CD, caesarean delivery.

study settings. To that end, the current study included referred mothers from local health centres for primary CDs only, which increases the risk of fetomaternal complications.

Our study also found that peripartum hysterectomy, the need for ICU admission and urethral/bowel injury were higher among mothers who underwent CD at the second stage of labour; however, the small number of those complications are insufficient to conclude.

On the other hand, we observed a significantly higher association between CD performed at the second stage of labour and adverse neonatal outcomes, including a fifth min Apgar score <7, the need for NICU admission and neonatal death than CD performed at the first stage of labour.

In agreement with our findings, previous studies[13 22] showed a significant association between the second stage of CD and neonatal morbidity, including an increased incidence of fifth min Apgar score <7 and higher NICU admission rates, but failed to show an association with neonatal death. In disagreement, our findings show that the CD performed at the second stage of labour is not only associated with neonatal morbidity but also with neonatal mortality, in line with other studies.[14 22 26]

On the contrary, some studies even observed the insignificant impact of CD performed at the second stage of labour on adverse neonatal outcomes.[15 17 23–25] These might indicate that the mode of delivery being emergency caesarean section, referred from other centres and the poor quality of neonatal ICUs in our study settings could justify the observed difference, suggesting the need for improving intensive neonatal care in resource-limited settings. Also, obstetric surgical access is limited in African countries and is responsible for increasing referrals to tertiary centres; when performed, it is often too late to reduce perinatal deaths and is associated with poorer neonatal outcomes, especially during emergency CD.[8]

Thus, providing equitable obstetric surgical access and strengthening obstetric surgical and intensive neonatal care in low-income countries need immediate attention to improve maternal and perinatal outcomes.[26–29]

### Strength and limitations
The strength of our study is that only referred mothers are included as a source population, which allows a comparison of the obstetric outcomes of these underserved populations to exemplify the existing problems. However, the retrospective nature of the study design precludes the inclusion of the most significant covariates, which might affect the impact of the stage of labour on the outcome variables. We also relied on data retrieved from the medical records of a single institution, making it difficult to extrapolate the findings to the non-referred mothers in the study setting. Furthermore, because of substandard overall NICU care and poor documentation in our hospital, we failed to include additional adverse neonatal outcomes, such as septicaemia, intubation, seizures and fetal injuries. Therefore, a further prospective study, including large sample size, multiple study hospitals and various neonatal outcomes, is needed to determine the generalisability of the findings to the entire study population. Despite these limitations, obstetric care providers should consider the impact of second-stage CDs and provide effective management to lower the burden of such cases.

### CONCLUSION
CS performed at the second stage of labour among referred mothers is associated with an increased risk of adverse maternal and neonatal outcomes. Thus, strengthening and improving obstetric emergency surgical services and intensive neonatal care for those populations would help reduce poor maternal and fetal outcomes. Further investigation may also require improving obstetric and neonatal intensive care.

**Acknowledgements** The authors acknowledge Wolkite University Specialized and Teaching Hospital Management, data collectors and medical record office staff for their invaluable support.

**Contributors** DZ, TT, FD and MA conceived and designed the study. DZ and TT performed the data collection, data analysis and initial draft manuscript. DZ and MA reviewed the manuscript critically. DZ as guarantor accepts full responsibility for the work and/or the conduct of the study, had access to the data, and controlled the decision to publish. All authors read and approved the final manuscript.

**Funding** The authors have not declared a specific grant for this research from any funding agency in the public, commercial or not-for-profit sectors.

**Competing interests** None declared.

**Patient and public involvement** Patients and/or the public were not involved in the design, or conduct, or reporting, or dissemination plans of this research.

**Patient consent for publication** Not applicable.

**Ethics approval** We have conducted the study per the Declaration of Helsinki Ethical Principles for Medical Research involving human subjects protocol. The study was approved by the Wolkite University Ethical Review Board and informed written consent was waived due to the retrospective nature of the study design.

**Provenance and peer review** Not commissioned; externally peer reviewed.

**Data availability statement** Data are available on reasonable request.

**ORCID iD**
Dereje Zewdu http://orcid.org/0000-0001-9819-1842

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
