## [Reviewer comments · BMJ Open]

ARTICLE DETAILS

TITLE (PROVISIONAL)	Association between the stage of labor during cesarean delivery with adverse maternal and neonatal outcomes among referred mothers to tertiary centers in resource-limited settings
AUTHORS	Zewdu, Dereje; Tantu, Temesgen; Degemu, Fikretsion; Abdlwehab, Mukerem

VERSION 1 – REVIEW

REVIEWER	Mekic, Nina Health and Educational Medical Center Tuzla, Pediatric
REVIEW RETURNED	28-Jul-2023

GENERAL COMMENTS	In Table 1, maternal and neonatal baseline characteristics should be marked as two separate entities. The study provides good insight into maternal outcome, but it would be of great value to include more adverse neonatal outcomes such as septicemia, need for intubation, neonatal seizures, fetal injuries...
---

REVIEWER	Pebolo, Francis Gulu University, Reproductive Health
REVIEW RETURNED	28-Aug-2023

GENERAL COMMENTS	I read your manuscript with lots of interests. This study present an important parts of obstetric care related to use of Caesarean delivery especially in low income countries. This said, I have the following comments for your attention: Abstract. The abstract looks good and succinct enough to provide a snapshot of the study. However, the Conclusion or recommendation seems you stated that strengthening referral systems with the local health enters would help to reduce this adverse outcome(Page 2, Lines 39-42). It is not clear in your study if there was any evidence that these outcomes was because of a weak referral systems. Line 17 page 2, please correct the spacing between 2023 and in Article summary Strength and limitations of Bullet #1 lines 9-12, page 3, the study being the first to be done in that population does not warrant it to have a strength. Maybe the methods, the sample size and the way Bullet #2, Lines 12-14, Page 3, that seems like a recommendation not a strength neither a weakness Introduction: No Comments Methods: Line 41-43, page 5 since the study only included primary CD, one would wonder how previous scar will become parts of the exclusion, since they are automatically excluded by nature of the study designs.
---

	Lines 50-54, page 5, under Data collection. you indicated that the data collection tools were used elsewhere would be good to give a citation(s) for the studies in which this was used. Lines 8-10, page 6, you indicated that one of the parameters you were getting from the files is peri-operative unintentional uterine extension. Did You mean unintentional uterine tear extension? If so, your obstetricians must be really honest to indicate that in the surgery notes. Its very unlikely in my settings for people even to mention that or classify that. Discussion section, Paragraph beginning page 11, lines 48 and page 12, lines 17 you have use the abbreviations CS, I supposed you its for Caesarean Section, Whille the rest of the texts, You have used CD for Caesarean Delivery. It would be good to stick to one. Conclusion, Same comments as the one in the abstract section.
--	---

VERSION 1 – AUTHOR RESPONSE

Reviewer 1

(I inserted line numbers to the manuscript to make review easier). Thank you for the opportunity to review this interesting work.

Thank you for your interest, and time to review our manuscript.

Line-N/a

Introduction

Line 56. SPPH not defined Thank you for your constructive comments. We clearly define the term severe postpartum hemorrhage in abstract and method section.

Line 136

Line 57. Could births replace deliveries? Thank you for your constructive comments. We replaced the term deliveries by births. Line 57

Line 59 severe PPH written despite SPPH being used previously.

Also used in lines 78, 79, 82 and throughout the whole text. If you are going to keep severe PPH then drop the SPPH and be consistent throughout. Thank you for your constructive comments. We have made corrections throughout the documents, including lines 78, 79, 82 and throughout the documents.

Line 70 CD used but no previous referral to this as caesarean delivery.

Thank you for your constructive comments. We have made corrections as suggested.

Line 70-71

Line 74-76 'However, it is unclear whether the available risk assessment tools can predict PPH in women who underwent CD, regardless of their mode of delivery'. Not sure what this means

There is assessment tools used to predict the risk of PPH; however, it's not specifically designed to predict severe PPH amongst women who underwent cesarean deliveries. Thereby, there is paucity of evidence whether those prediction tools could be useful in these scenarios.

We made it clear by expressing the main point using other words. Line 74-76

Line 83 states "severe PPH may also occur in women with no known risk factors (4)". Thank you for your constructive comments. We corrected as PPH instead of severe PPH.

Line 83

But this is the same for PPH, and there is no reference here to the fact that a SPPH could be a poorly managed PPH, or a well-managed PPH could prevent SPPH.

Thank you for your constructive comments. Concerning the reference to the fact that a severe PPH could be a poorly managed PPH or a well-managed PPH could prevent severe PPH; we apparently described it in a well manner.

Line 59

Line 84. This paragraph needs rewording, as currently is not clear.

Thank you for your constructive comments. We made rewording as suggested.

Line 84

Line 85 after (29) requires a colon rather than a full stop, as this changes the sense of these important messages. Thank you for your constructive comments. We made correction as suggested.

Line 85

Line 91. Refers to existing variation needing consideration in light of potential differences according to mode of birth. But this has not been considered previously. Neither has the fact that some countries use different definitions of PPH for CS or VB and the impact of this on PPH rates and progression to SPPH. Also seems to be referring to PPH at the start of this para, not SPPH. Thank you for your constructive comments. We made correction as PPH instead of severe PPH as suggested.

Line 91

Line 92 should start with In spite, rather than spite. Thank you for your constructive comments. We made correction of typo error.

Line 92

Line 95. "modifiable risk factors can be intervened" is ambitious, given that many of the woman have no risk factors, presumably apart from emergency CS in labour. Thank you for your constructive comments. According to study, not many women but some have no risk factors; thus it allows the clinicians to intervene on the known modifiable risk factor, that's what we mean.

Line 95-96

Line 102 aims to "reduce the prevalence of PPH after caesarean delivery" but this is not the same as SPPH.

Thank you for your constructive comments. We made it to reduce the prevalence of severe PPH instead of PPH as suggested.

Line 102

Methods

Line 107. Does this refer to any missing documentation?

Thank you for constructive comments. Of course it also refers to any missing documentation.

Line 107

Line 120. H/O abortion ever? Abnormal uterine bleeding, and single/multiple pregnancy presumably in current pregnancy? Thank you for your constructive comments. We made it clear as follows; Previous history of abortion and abnormal uterine bleeding, current singleton or twin pregnancy.

Line 120

Line 121. What is prepartum anaemia? Is this a history of anaemia outside pregnancy, or is it booking HB/ferritin levels? Presumably severe pre-eclampsia (? How defined) HELLP, APH (would this not be covered by abnormal uterine bleeding) in current pregnancy. Thank you for your constructive comments. We tried to look at their definitions under operational definitions.

APH is not covered by abnormal uterine bleeding, and prepartum anemia also not includes history of anemia outside pregnancy.

Line 141.....

Line 121 should it not be HELLP syndrome, not HEELP as written?

Thank you for your constructive comments. We made corrections as suggested.

Line 121

Line 118-125. I would suggest you group these as pre-pregnancy, pregnancy acquired, intrapartum factors. This would make this section easier to follow.

Thank you for your suggestions. We made revisions in a suggested way of grouping.

Line 118-123

Line 129. Would be clearer if you break down prophylaxis and treatment especially if you use misoprostal prophylactically, as many places don't, so this would be of interest to readers in other jurisdictions.

Thank you for your constructive comments. We made corrections by breaking down the prevention and treatment of PPH as suggested.

Lines 128, 131 & 133-139

Line 131. I'm not sure whether in actual fact only oxytocin is used for prophylaxis, in which case the sentence above needs tweaking.

Thank you for your constructive comments. We made revisions concerning prophylaxis and treatment approach of PPH in our setting.

Line 134-136

Line 134. When was baseline? Early in labour/prior to CS or earlier in pregnancy?

Thank you for your constructive comments. We define baseline hemoglobin: the last measurement of hemoglobin level prior to cesarean section, and included under operational definitions.

Line 145

How would the clinician evaluate- as you relying on visualisation or measurement, or maternal symptoms of anaemia? Thank you for your constructive comments. As the retrospective nature of study design we relied on measurement than clinical observations. The narration of diagnosis methods has been used to describe the trends in our institution. For the record, we clearly describe how we diagnose

severe PPH.

Line 141-144

Results

Would be interesting to see how many women were excluded due to incomplete data. Therefore what proportion of total CS births were the 728 included women? Thank you for your constructive comments. We included the participants excluded due to incomplete data as suggested.

Line 110-114 & 174-175

Line 155. Presumably the 2 groups were those with, and those without SPPH? Please explain in text. Please define poor antenatal follow up- how many visits, reasons for non-compliance etc etc. Thank you for your constructive comments. We made corrections as suggested. However, as this study is not aimed to investigate the reasons for non-compliance rate, it is difficult to discuss here.

Line 176-178

Line 161. Remove s from obstetrics Thank you for your constructive comments. We made corrections as suggested.

Lines 183-84

Line 164. who are "their counterparts"? Thank you for your constructive comments. In this context, it is obviously clear that their counterparts are those women who presented with 2 or more CS history, APH and severe preeclampsia.

Line 187

Table 2: presumably preterm was 28+1- 37+0 but this needs to be defined. Thank you for your constructive comments. We included it under operational definitions.

Line 160

Failed instrumental delivery, could this be relabelled failed instrumental vaginal birth Thank you for your constructive comments. We made corrections as suggested.

Line 176 & 188

Malpresentation- please define. Does this include Breech? If yes, might be work separating as a non-cephalic presentation. Thank you for your constructive comments. We made corrections as suggested.

Line 188

Discussion

Line 197. Again, it would be useful to see how this 728 is representative of the overall population. Could these 728 introduce bias? Thank you for your constructive comments. We included the number of participants who were excluded because of incomplete documentation. To avoid repetitions, we included it in method and result sections.

Lines 111-113 & 174-175

Line 198. Is it appropriate to compare the Ethiopia with the US? Due to the complexity of the health system in the US, comparison is often difficult. What was the rationale for inclusion of these countries? Are the health systems comparable?

Thank you for your constructive comments. The main reason comparing our results with the US is because of the paucity of studies investigating severe PPH among intrapartum cesarean deliveries. We also clearly point out the justifications for the observed differences including the health system.

Line 234-236

Line 206. It has been argued that number of units transfused is more reflective of health service policies, resource availability and clinician management than severity of bleed. This could be particularly the case here as the authors have alluded to lack of availability of some blood products.

Thank you for your constructive comments. As I mentioned earlier we determine the severity of bleeding based on the derangement of hemoglobin measurement than clinical observation because of retrospective nature of the study design. We also clearly state management protocols of the institution.

Line 143-146

Line 208. It has also been argued that estimating blood loss at CS is more accurate than other modes of birth, due to the weighing of all swabs and dressings routinely undertaken in theatre.

Thank you for your constructive comments. Unsurprisingly, estimating blood loss during CS is relatively more accurate; but the challenging issues have been aroused because of underestimation by obstetricians as confirmed by many studies.

That's why we relied on hemoglobin measurement, amount of blood transfusion and hemostatic interventions than underestimated blood loss for this study.

Lines 144-146

Line 225 states that women with severe pre-eclampsia and eclampsia, but previously eclampsia was not mentioned, rather HELLP syndrome. Were there no women in the study with HELLP? Subsequently in Line 227 the authors appear to suggest that severe pre-eclampsia and eclampsia resulting in HELLP syndrome cause coagulopathies leading to PPH. But PE has long been identified as a risk factor for PPH in the absence of HELLP.

Thank you for your constructive comments. We made corrections as suggested. The number of HELLP syndrome and eclampsia were insignificant, that why we left to include.

Line 250-252

Line 230. Were the mothers with 2 or more previous CS births at increased risk due to abnormal placentation (praevia, accreta, increta, percreta) associated with previous uterine scarring? This is not covered in the potential explanation of uterine atony or adhesions.

Thank you for your constructive insights. We included the increased risk of abnormal placentation has an impact in PPH.

Line 258

Line 245. Would be interesting to discuss the impact of GA versus local anaesthesia in light of practice in your service. Commonly these days even the sickest mothers tend to have CS under regional anaesthesia, and yet the PPH rate continues to rise.

Thank you for your constructive comments. We believe that to comment the impact of GA vs regional anaesthesia in PPH need further investigation; otherwise without controlling the confounding factors it could be difficult for discussion.

Line 264-265

Line 255. Again, what proportion of women in your service would have a classical incision? In the UK and Australia a classical incision is commonly only used with an extremely preterm infant, which would be excluded from your study (<28 weeks). You also mention CS in labour and worth the growing concerns around full cervical dilatation CS and subsequent mid- trimester pregnancy loss or PTB, utilisation of a higher incision at this time could be topical and improve subsequent pregnancy outcomes and could be a great discussion point. This also might be more significant than proposing further trials re use of classical incisions. Thank you for your constructive comments. We made corrections as suggested.

As we described it in result section, only 20/728 women were underwent classic incision.

Likewise, by removing the statement says "future studies are needed to determine this matter", we included the importance of adapting guidelines that tailored to reduce classical incision.

Line 284-286

Line 268. Rather than simply the inclusion of anaesthesiologists should you not be advocated for a multidisciplinary highly skilled team? Thank you for your constructive comments. We made corrections as suggested.

Line 292-293

Line 273. Describes the value of using pre and post Hb level comparisons and requirement for blood, but these only work in retrospect. So whilst reliable, they are of limited use in managing the situation as you need to wait for the blood test results to be available. Thank you for your good insights.

We included this critical point in the limitation. Using the value of hemoglobin difference before and after CS and requirement for blood transfusion might delay the diagnosis of severe PPH as the clinicians need to wait the blood test results.

Line 279. States the limitation of the study was that known risk factors such as pregnancy interval and previous H/O PPH were not included in the analysis and yet in the results it states that previous PPH was a contributory factor. Or were these women all excluded from the analyses, and therefore there were an additional 437 women who underwent CS but were excluded from this analyses. Or were all 728 women included but there was missing data for 437, in which case the analyses were on 291 cases? Thank you for your constructive comments. We made corrections as suggested. We are saying that women (>60%) were referred from other institutions where their previous medical records are unknown.

We didn't say previous history of PPH was a contributory factor in the result section.

Line 290

Conclusion

Line 284 You state that the incidence of SPPH after CS was relatively high, and yet you have shown that it is not dissimilar to other published literature. You identify no new findings regarding risk factors to those already published in the literature. There is no new information around how to identify predictors or for policymakers to identify screening tools.

Thank you for your constructive comments. We revised as it's relatively similar with other studies. And

also we state recommendation based on our finding.

As far as our knowledge, this paper is among few studies focused on severe PPH following cesarean deliveries and contains plenty of new information. We have revised the conclusion and recommendation for stakeholders.

Line 311-315

General

Caesarean delivery should be changed to caesarean section or caesarean birth. There are several typos throughout the text.

Thank you for your constructive comments. We made corrections throughout the documents.

I think the discussion could be presented more effectively. The authors talk about strategies to prevent and manage PPH, they also state that TXA and certain blood products are unavailable in their service, I would have liked to see more information about whether these data could be used to support introduction of TXA particularly, given it is cheap and requires no special storage.

There are trials of introducing different blood products and tranexamic acid into our setting, but the existing bureaucracy challenges its translation into clinical practice. We will receive this suggestion even to evaluate the changes in the overall rate of severe PPH after introducing different blood products and tranexamic acid into our setting.

Line-N/a

References

Formatting of references not consistent throughout document. Some have only primary author, et al, others have full list of authors. Some have date after authors, other have it after name of journal
Thank you for your constructive comments. We made corrections as suggested.

References

Khan's review of causes of maternal death is now 17 years old with data predating that. Is there nothing more contemporary available?

Thank you for your constructive comments. We replaced it with other reference as suggested
Some of the other references are similarly quite old- is there any more contemporaneous evidence in the literature, given that the demography of women (age, BMI), service configuration (changes in doctors hours, scope of midwifery practice) and prevention and management of PPH (TXA,) have all been modified in recent years?

Thank you for your constructive comments. We replaced it with other reference as suggested.

My recommendation would be MAJOR REVISION. The reasons for this are:

Throughout the text it states severe PPH and PPH interchangeably. I am not always sure which you are referring to.

Thank you for your time, consideration and constructive comments; it helped us to improve our manuscript in many ways.

We made corrections and proof readings throughout the documents.

Line- throughout the documents

The methods require more detail, for example regarding when the first blood test is taken for comparison and therefore diagnosis of SPPH, it states baseline, but when is that?

Thank you for your constructive comments. We clearly described it under operation definition section.

Line 154-156

The stats are confusing, and could be more simply presented.

Thank you for your constructive comments. We made critical revisions as suggested.

Line 164-170

The data collection points are not logically presented.

Thank you for your constructive comments. We critically revised the data collection points as suggested

Lines 136-147 & 153-160

The discussion is weak and could be strengthened, given the available data.. The strengths and limitations are not well describes, indeed the strength could be considered a weakness, as reliance of blood results can cause delay in treatment, and therefore this measurement of blood loss is only useful retrospectively. the work does not support the conclusions. Thank you for your constructive comments.

We tried our best to strengthen the discussion including the strength and limitations in a logical ways.

Lines 222-225, 233-235 & 250-251

Reviewer 2

Comments for PPH after CD

1. This is a good study that will assist policymakers in setting strategies for decreasing maternal death. If the authors revise the manuscript carefully, it will be a good asset to the scientific world. My recommendation is a major revision.

Thank you for your time, consideration and constructive comments to improve our manuscript.

Line- N/a

2. In your title, the study was conducted at Wolkite, why do you prefer to say Ethiopia? Is it representative? Make it specific.

Thank you for your constructive comments. As our study area is a referral center for South Central Ethiopia, We made corrections as suggested.

Lines 2, 98, 207

3. You need to avoid abbreviations in the abstract. E.g., PPH, APH....

Thank you for your constructive comments. We have removed out every abbreviations used in the abstract section.

Lines 26, 29, and throughout the abstract

4. In the method of analysis, why do you prefer logistic regression? It is better to consider other models for this cohort study?

Thank you for your constructive comments. There is no ideal method of analysis than logistic regression to fully address the objectives of our study.

Line - N/A

5. It is better to report the incidence rate? Prevalence is common for cross-sectional studies.

Thank you for your constructive comments. We made corrections throughout the documents, including title, introduction, results and discussion.

Line- throughout the documents

6. Your conclusion should be revised. It is not good.

Thank you for your constructive comments. We made corrections as suggested.

Lines 46, 47 & 49

7. Your introduction section is disorganized and too long. It is better to focus on the burden of PPH, risk factors, consequences of PPH, tried efforts, and current recommendations and gaps of the study. Please rewrite based on those sequences.

Thank you for your constructive comments. We have tried our best to re-organize and minimize the introduction sections as suggested.

Lines 65-66, 72-73, 79

8. Please avoid the use of abbreviations at their first appearance.

Thank you for your constructive comments. We made corrections throughout the documents.

Lines 54, 65-66 & 68-70

9. In the method section: you merged different ideas in one sub-topic. You need to separate it.

Thank you for your constructive comments. We made corrections as suggested.

Lines 137 & 143

10. Your method section is incomplete? Sampling techniques? Sample size determination? etc.....

Thank you for your constructive comments. We already described it in line 103-104, but to avoid further ambiguity we have created a sub-heading.

Line 110-112

11. Is your tool validated? Is your tool reliable?

Thank you for your constructive comments. Of course we had used a standardized checklist that adapted from previous literature and pre-test was carried out on 5% of sample size outside study setting.

Line 116-117

12. Better to revise your statistical analysis that fits your design.

Thank you for your constructive comments. We made corrections as suggested.

Line 162-167

13. Since your study is retrospective, medical card incompleteness is a major issue in majority of health settings? How is your study site different? How do you retrieve all of the samples from medical records?

Thank you for your constructive comments. Data extraction was performed independently by two trained data collectors from all medical records using standardized pretested checklist adapted from previous

studies. We also discuss this problem as a limitation of study.

Line 116

14. Your result lacks subheadings.

Thank you for your constructive comments. We made corrections as suggested.

Lines 170, 180, 188 and 197

15. Do not narrate everything presented in the tables.

Thank you for your constructive comments. We tried our best to minimize it throughout the result sections.

Line – N/a

16. In your final model, you should present two-by-two tables.

Thank you for your constructive comments. We made correction to present the final model with two-by-two table.

Line 217

17. The discussion section should be revised extensively. There is no reference for each idea. There is the implication of each finding

Thank you for your constructive comments. We made an extensive revision throughout the discussion section including limitation and strength of the study as suggested.

18. Conclusion should be strong and revise it based on your findings.

Thank you for your constructive comments. We made critical revision to present the conclusion section based on findings.

Line 313-318

19. Include the ethical clearance number.

Thank you for your constructive comments.

We included the ethical clearance number as suggested. RCSUILC/55/22

Lines 108 & 331-332

20. Revise the whole document for typos and grammar errors.

Thank you for your constructive comments. We made revisions for type setting and grammar for the whole documents.

Line- throughout the whole documents

VERSION 2 – REVIEW

REVIEWER	Mekic, Nina Health and Educational Medical Center Tuzla, Pediatric
REVIEW RETURNED	31-Oct-2023

GENERAL COMMENTS	The basic characteristics of the mother and the newborn are marked as two separate entities, as requested. The lack of adverse neonatal outcomes such as septicemia, need for intubation, neonatal seizures, fetal injury is discussed in "Strengths and Limitations". I still believe that additional adverse neonatal outcomes would add to the value and significance of the study, but even in the absence of such data, the study provides good insight into adverse maternal outcomes.
--

REVIEWER	Pebolo, Francis Gulu University, Reproductive Health
REVIEW RETURNED	29-Oct-2023

GENERAL COMMENTS	All the comments had been adequately addressed. Just maybe one or two minor Page 2 line 37, I have seen Caesarean section yet in most we have used Caesarean Delivery. would be good to stick to one? I am not so sure.
---

	A careful read of strengths and weaknesses/limitations in lines 52-59 and lines 257-269 means that these two sections need to be compared and the article summary section should be a summary of what is in lines 257-269.
--	--